# Template-Assisted Assembly of Hybrid DNA/RNA Nanostructures Using Branched Oligodeoxy- and Oligoribonucleotides

**DOI:** 10.3390/ijms242115978

**Published:** 2023-11-05

**Authors:** Alesya Fokina, Yulia Poletaeva, Svetlana Dukova, Kristina Klabenkova, Zinaida Rad’kova, Anastasia Bakulina, Timofei Zatsepin, Elena Ryabchikova, Dmitry Stetsenko

**Affiliations:** 1Faculty of Physics, Novosibirsk State University, Novosibirsk 630090, Russia; a.fokina@nsu.ru (A.F.); k.klabenkova@g.nsu.ru (K.K.); 2Institute of Cytology and Genetics, Siberian Branch of the Russian Academy of Sciences, Novosibirsk 630090, Russia; 3Institute of Chemical Biology and Fundamental Medicine, Siberian Branch of the Russian Academy of Sciences, Novosibirsk 630090, Russia; fabaceae@yandex.ru (Y.P.); lenryab@niboch.nsc.ru (E.R.); 4JSC Genterra, Moscow 129085, Russia; kurakina_svetlan@mail.ru; 5Faculty of Natural Sciences, Novosibirsk State University, Novosibirsk 630090, Russia; 79513891932@yandex.ru (Z.R.); bakulina@gmail.com (A.B.); 6Department of Chemistry, Lomonosov Moscow State University, Moscow 119991, Russia; tsz@yandex.ru

**Keywords:** nucleic acids, DNA nanotechnology, solid-phase synthesis, branched oligonucleotide, photolabile protecting group

## Abstract

A template-assisted assembly approach to a C_24_ fullerene-like double-stranded DNA polyhedral shell is proposed. The assembly employed a supramolecular oligonucleotide dendrimer as a 3D template that was obtained via the hybridization of siRNA strands and a single-stranded DNA oligonucleotide joined to three- or four-way branched junctions. A four-way branched oligonucleotide building block (a *starlet*) was designed for the assembly of the shell composed of three identical self-complementary DNA single strands and a single RNA strand for hybridization to the DNA oligonucleotides of the template. To prevent premature auto-hybridization of the self-complementary oligonucleotides in the starlet, a photolabile protecting group was introduced via the *N*^3^-substituted thymidine phosphoramidite. Cleavable linkers such as a disulfide linkage, RNase A sensitive triribonucleotides, and di- and trideoxynucleotides were incorporated into the starlet and template at specific points to guide the post-assembly disconnection of the shell from the template, and enzymatic disassembly of the template and the shell in biological media. At the same time, siRNA strands were modified with 2′-OMe ribonucleotides and phosphorothioate groups in certain positions to stabilize toward enzymatic digestion. We report herein a solid-phase synthesis of branched oligodeoxy and oligoribonucleotide building blocks for the DNA/RNA dendritic template and the branched DNA starlet for a template-assisted construction of a C_24_ fullerene-like DNA shell after initial molecular modeling, followed by the assembly of the shell around the DNA-coated RNA dendritic template, and visualization of the resulting nanostructure by transmission electron microscopy.

## 1. Introduction

One of the most dynamic and fast-developing areas of nucleic acid chemistry is DNA nanotechnology [1,2], which employs DNA molecules as a unique material for diverse 2D or 3D nanoarchitectures [3,4], which can be rationally designed and precisely controlled to furnish the components of dynamic DNA nanodevices [5,6,7], artificial nanocompartments [8], nanomachines [9,10], and nanorobots [11,12,13]. At the very foundation of this area of science lies the intrinsic propensity of polynucleotides to form double-helical complexes via complementary Watson–Crick base-pairing, as well as self-assemble into higher-order structures such as triplexes, G-quadruplexes, or i-motifs [14,15,16]. In 2006, an attractive method of DNA origami was proposed [17], which involved the folding of a long single-stranded DNA template mediated by a set of oligonucleotide “staples”, each complementary to a specific DNA sequence within the template, into a predetermined 2D or a 3D shape [18,19,20]. Over the past 15 years, the method has grown into a versatile, practical technology for creating functional DNA nanostructures with many uses, including bioanalysis [21], biosensing [22,23,24], bioimaging [24,25,26], nanophotonics [27,28], cancer therapy [28], immunotherapy [29,30], gene therapy [31], tissue engineering [32], and drug delivery [33,34]. The last application of DNA-based nanoparticles appears to be among the most promising, particularly for the delivery of antisense oligonucleotide therapeutics into bacteria to combat the spreading of antibiotic resistance [35,36]; also see ref. [37] and references cited therein. Most of the modern DNA origami techniques are based on a combination of computer algorithms for programmable self-assembly of DNA molecules [18,19,20] with methods of molecular biology for manipulating large DNA molecules [38]. More and more attention are being paid to chemical modification of nucleic acid building blocks in the rational design of nanoscale architectures, especially for therapeutic applications [39,40,41]. Site-specific chemical modification by extraneous groups may affect interactions of DNA nanostructures with living cells [42], including cellular uptake [43,44], and prevent their intracellular and in vivo degradation [45]. Therefore, supplementing DNA origami approaches with those based on solid-phase oligonucleotide synthesis and chemical modification/functionalization of nucleic acids in order to create new technologies for controlled assembly of 3D nanoobjects and functional nanodevices remains a highly relevant task.

Previously, we proposed a rational approach for constructing DNA polyhedra via a template-assisted assembly that employs branched oligonucleotidic building blocks to construct the apices and edges of a specific DNA polyhedron (a *DNAhedron*). The task can be achieved by hybridization to a specifically formed branched oligonucleotidic template serving as an internal 3D scaffold to support the shape of the target polyhedron, e.g., tetrahedron, cube, octahedron, etc. [46]. The method of choice for obtaining the corresponding branched oligonucleotides seemed to be a combination of automated solid-phase synthesis using, apart from nucleoside phosphoramidites, non-nucleosidic phosphoramidites of specific functionality, such as doublers, treblers, or cleavable linkers, followed by either post-synthetic ligation in solution, e.g., via click chemistry, or a Watson–Crick hybridization to form supramolecular noncovalent assemblies.

Initially, we have justified via calculations and molecular modeling a set of oligonucleotide sequences for the branched oligonucleotidic templates for the assembly of a DNA tetrahedron, a DNA cube, and a DNA hollow shell analogous to C_24_ fullerene. Simultaneously, a 4-way junction oligonucleotidic building block (nicknamed by us as *starlet*) common to all three above topologies was selected [46]. Commercially available non-nucleosidic branching phosphoramidites (symmetric doubler and trebler) and modifying phosphoramidites (e.g., C6 disulfide linker) were employed for the synthesis of the templates and the starlet. Next, branched oligonucleotide components for the corresponding oligonucleotidic templates for the assembly of a DNA tetrahedron and a DNA cube, and the starlet that was common for both *DNAhedra* were synthesized. Finally, an assembly of a DNA tetrahedron from the starlet units assisted by the relevant branched template was carried out, and the outcome was checked by transmission electron microscopy [47].

In this short communication, we describe the synthesis of a novel set of branched oligoribonucleotides and mixed oligodeoxy-/oligoribonucleotides (see Figure 1) for a template-assisted assembly of a DNA hollow shell topologically related to C_24_ fullerene, the assembly of the shell on a branched DNA/RNA template (see Figure 2), and transmission electron microscopy visualization of the resulting nanostructures.

## 2. Results

The C_24_ fullerene, of which two topological isomers exist (C_24_ (*O_h_*) and C_24_ (*D_6_*), respectively [48]), was previously considered for drug delivery applications [49,50]. We have designed a C_24_ fullerene-like hollow DNA shell with 24 vertices introduced by 24 starlets **F** (Figure 1), each vertex corresponding to the 4-way junction of **F**. The edges of the polyhedron were formed by antiparallel partially self-complementary duplexes from oligodeoxynucleotides ODN8 (Figure 1). The respective oligonucleotidic template for the assembly of the C_24_ shell was decorated on the outside with 24 DNA single strands ODN1 to hybridize to the 4th branch ORN7 of each of the 24 units of the starlet **F** (Figure 1). It was proposed to assemble such a 24-valent template from a set of branched oligonucleotides including a 3-way junction oligonucleotide **A** having two DNA branches ODN1 and one RNA branch ORN2, and oligoribonucleotides **B** (3-way junction) and **C** (4-way junction), with the formation of a supramolecular, i.e., having no covalent bonds between the constituent oligonucleotides, dendron **D** via hybridization of the respective complementary single strands. The next stage involved dimerization of the resulting dendron **D** into the full-size template **E** via self-complementary duplex **6**:**6** formation after the removal of the photolabile NPOM group from ORN6 by UV irradiation at 365 nm (Figure 2). Such a convergent assembly of template **E** was expected to decrease spatial difficulties in the formation of the outer layer of 24 single strands ODN1 and reduce the proportion of defective dendrons (polydispersity).

The free, open-source software Blender (version 2.82 was used in this work) [51] was widely employed for a variety of 3D modeling and visualization tasks in molecular biology, e.g., for visualization of antibodies [52], for protein design [53], for a 3D representation of a DNA origami receptor [54], and for modeling of amyloid fibrils [55]. Using Blender, relatively simple yet informative models of DNA tetrahedron and DNA cube were previously obtained with the selected parameters [47], followed by a more complex model of a C24 fullerene-like DNA shell in the current work (Figure 3). It turned out that Blender is well suited for the visualization of nanostructures such as these and for selecting parameters that allow for their assembly without obvious spatial strain [46,47].

We previously hypothesized [46] that the complex of the DNA shell with the hybrid DNA/RNA template may serve as a delayed action RNA interference (RNAi) agent if the internal duplexes of the template **2**:**3** and **4**:**5** (Figure 2) were constructed as small interfering RNAs (siRNAs). It was shown that some branched RNAs showed biological activity according to RNAi mechanism [56]. Sequences of siRNAs targeting gankyrin [57] and PARP-1 [58] were used in the calculations and subsequent synthesis of branched oligonucleotides **A**, **B**, and **C** (Table 1). To increase the enzymatic resistance of the siRNAs, some of the ribonucleotides were replaced by 2′-*O*-methylribonucleotides (Table 1). Furthermore, in all oligonucleotides, the two phosphodiester groups at the 5′-termini and in the siRNAs, additionally, one more phosphodiester nearest to the 3′-terminal cleavable unit (either di-2′-deoxynucleotide or tri-ribonucleotide) were replaced by phosphorothioate groups (Figure 1, Table 1). We assumed that the presence of 2′-*O*-methylribonucleotide and phosphorothioate modifications would not significantly affect the simulation results.

To facilitate the hydrolytic fragmentation of the branched template into individual siRNA duplexes after enzymatic cleavage of the shell, which was expected predominantly at the trinucleotidic 5′-d(TCT) single-stranded sites adjacent to **1**:**8** duplexes of the shell, it was decided to introduce into the interior of the template additional triribonucleotides 5′-r(UAC), which are a known to be a good substrate for ribonuclease A (Figure 1).

The data from the paper [59] were used to design the oligonucleotide sequences and evaluate the thermodynamic stability of the duplexes therefrom. A Blender-generated image of a simulation result for a DNA hollow shell, whose topology corresponds to C_24_ (*D_6_*) fullerene [48], in a complex with 24-valent template **E** is shown in Figure 3. The respective DNA/DNA duplexes (in the canonical B-form) and RNA/RNA duplexes (in the A-form) are represented as colored cylinders connected by flexible linkers shown as chains.

The synthesis of all the branched oligonucleotides was accomplished by solid-phase phosphoramidite chemistry on a commercial automated DNA/RNA synthesizer, followed by isolation and purification by gel electrophoresis (see Section 4). The identity of the oligonucleotides was confirmed using mass spectrometry (see Appendix A).
Figure 3An optimized configuration of a C24 fullerene-like DNA shell (transparent green) in a complex with the 24-valent template **E** as created in Blender software. Duplexes are represented as colored cylinders connected by linkers in the form of chains. The coloring loosely follows Figure 1 (including **F**): transparent green—self-complementary duplexes **8**:**8** (the shell); solid blue—duplexes **1**:**7**; solid yellow—siRNA duplexes **2**:**3** (msGankyrin-4, Table 1); solid red—siRNA duplexes **4**:**5** (msPARP-1-4, Table 1); solid cyan—a self-complementary central duplex **6**:**6**. DNA duplexes **8**:**8** of the shell are placed inside transparent green cylinders as canonical B-form double helices [59].
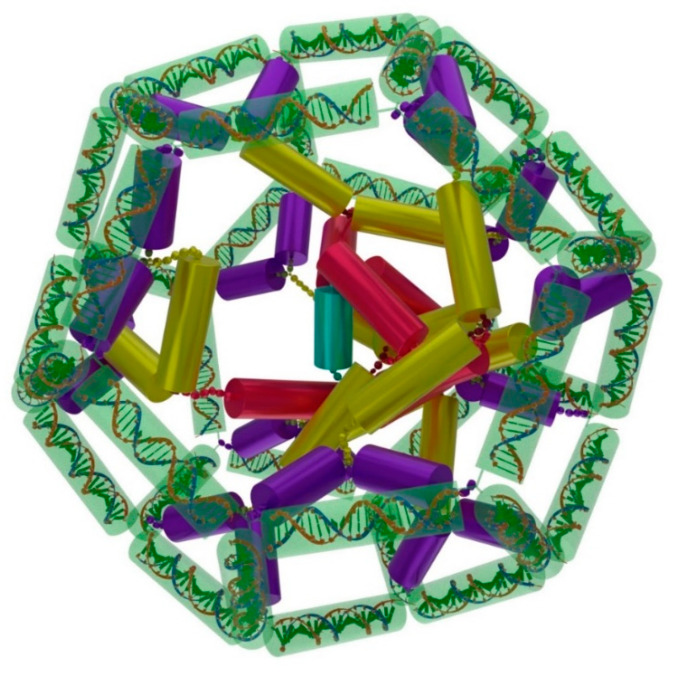


The assembly of the half-template dendron **D** via hybridization from branched oligonucleotides **A**, **B**, and **C** (Figure 2) in the stoichiometric ratio of 6:3:1, respectively, was carried out either on a water bath or in a dry thermostat. Significantly different TEM results were later observed for these two techniques. The process continued via the dimerization of **D** into the 24-valent template **E** after 5 h of UV irradiation at 365 nm (see Section 4).

Examination via TEM of the samples of **E** and **F** (1:24 ratio) after the assembly on a water bath revealed discrete objects roughly spherical in shape with a diameter of ca. 35 nm. The objects were composed of “rods” of ca. 2 nm thickness, forming a hexagon (Figure 4B–E, white arrows). The observed shape of the nanostructures corresponds to a polyhedral shell topologically related to C_24_ (*D_6_*) fullerene [48] spread out on the substrate on the wider (hexagonal) side and seen from the top (see Figure 3). Inside the hexagons, a core-like element was visible, which had a higher electron density than the shell, which may correspond to a shrunk 24-valent RNA template **E**. These cores had the form of a ring ca. 10–15 nm in size with transverse striation (Figure 4B–E, black arrows). Depending on how the particle was adsorbed and dried, one or another of its elements was better seen in the images: “rods”, cores, or side faces. In the samples, particles were also observed that lie independently of the shell-like objects, similar in size and shape to the isolated cores (Figure 4J–L). These core-like objects may correspond to either individual templates **E** or truncated shells. In addition to discrete objects, the sample contains an unformed medium electron density material, probably other synthesis components.

Assembly of the same nanostructure in a dry thermostat, according to TEM, was much less efficient. The sample contained only a few discrete objects, ca. 35 nm in size. However, unlike the above-described shell-and-core-like objects, their structure was fuzzy, and the cores were poorly visualized or not visualized at all (Figure 4F–I). Separate rounded objects similar to the cores and having ca. 10–15 nm in size were located on the substrate singly (Figure 4M–O); the details of the structure were not visualized at all. The unformed material of medium electron density was more structured than in the sample obtained in a water bath.

Attempted assembly with the control non-complementary starlet **F*** on a water bath (Appendix A) and in a dry thermostat (Appendix A), according to TEM, resulted only in singular objects with a definable structure, roughly hexagonal or oval in shape, with a central rounded element. Also, rounded particles ca. 10–15 nm in size lying separately on the substrate, probably corresponding to the isolated templates **E**, were observed (Appendix A). The main part of the sample consists of unformed medium electron density material, which is more structured in a dry thermostat.

## 3. Discussion

The project was inspired by a concept of a nanorobot constructed of hybrid DNA/RNA nanoarchitectures and aimed at carrying out medicinal tasks inside the body such as, e.g., targeted drug delivery. The main structural features of such a nanorobot are a hollow DNA shell, which is a nucleic acid equivalent of a proteinaceous viral capsid, and a branched DNA/RNA nanostructure located inside the hollow shell, which includes biologically active double-stranded RNA such as small interfering RNAs (siRNAs), which can act as a drug load, or a “cargo”, of the nanorobot analogously to the viral genome. Thus, the internal nanostructure may not only guide the assembly of the outer DNA shell but may also be utilized as a functional load of the DNA-coated nanoparticle if it can be released as a result of enzymatic shell disintegration at specific cleavable positions.

We set out to demonstrate that, as a relatively simple model, a C_24_ fullerene-like hollow DNA shell may be constructed by a template-assisted assembly from 24 standard pieces (starlets), each of them containing a 4-way junction with three self-complementary single-stranded DNA branches of the same sequence, and a 4th branch complementary to one of the 24 sticky ends of a specially constructed as a 3D scaffold branched oligonucleotidic template. The key feature is the incorporation of a nucleotide masked by a light-removable protecting group, such as commercially available *N*^3^-NPOM-dT, to prevent premature auto-hybridization of the self-complementary branches of the starlet until after hybridization to the template (Figure 1**F**). It was found that at least three of the masked nucleotides have to be introduced per 20 nt to block any undesired auto-hybridization. Examples of such a “caged” chemistry were shown previously by A. Deiters and coworkers [60,61,62].

At the initial stage of the project, computer modeling of the complex of a C_24_ fullerene-like hollow DNA shell with a branched RNA template was carried out using Blender software [47], which made it possible to optimize the geometric parameters and sequences of the DNA/DNA, DNA/RNA, and RNA/RNA duplexes from branched oligonucleotides that make up the template and the shell toward minimization of spatial strain in the resulting structure, and give recommendations for chemical synthesis.

In the current work, automated solid-phase synthesis, isolation, purification, and confirmation of the structure of the branched oligonucleotides **A**–**C**, and **F** (Figure 1) were carried out. At a given level of complexity and a set of chemical modifications, this step was not a trivial task and required special conditions for coupling of specialty monomers such as doubler, trebler, C6 S-S modifier, and “reversed” phosphoramidites for DNA or RNA synthesis in the 5′-3′ direction (see Section 4).

Next, a branched DNA/RNA template was assembled via hybridization (see Figure 2), followed by a hollow DNA shell on top of the template. The key step was a UV light-promoted removal of the NPOM protecting group from the single dT residue in the middle of the ORN6 sequence in the dendron **D**, optimized previously in the studies on template-assisted assembly of a DNA tetrahedron [47]. The same procedure was applied for the unmasking of the NPOM-dT residues in the branches of the starlet to initiate the formation of the DNA shell. At that stage, we detected a sharp increase in the fluorescence of the intercalating dye Ethidium Bromide, which was consistent with the formation of newly made B-type DNA duplexes (Appendix A, lanes 3, 4, 7, and 8).

Visualization of the constructed DNA/RNA nanostructure was obtained by transmission electron microscopy (TEM). It was shown that the use of an optimized technique for preparing and contrasting samples of nanostructures made it possible to successfully visualize the complex of a C_24_ fullerene-like DNA shell with a DNA/RNA template and draw conclusions regarding its topology.

As the angles between the DNA duplexes in a C_24_ fullerene-like DNA shell do not have strict constraints, unlike the angles between C-C bonds in true fullerenes, one can expect to obtain a variety of the resulting configurations of the DNA/RNA nanostructure in the modeling, and we can potentially see multiple topologies to form in the experiments. However, we think that both the modeling studies (Figure 3) and the TEM images (Figure 4B–E) seem to support the formation of a nanostructure topologically similar to the C_24_ (*D_6_*) fullerene isomer [48].

Although our modeling did not take into account the flexibility of the DNA and RNA duplexes, it proved to be useful for the estimation of basic geometrical parameters and generating structural input for chemical synthesis, as was confirmed by the agreement with the TEM data obtained. In the experiment with the complementary starlet **F**, the TEM images revealed numerous discrete nanostructures of the size of ca. 35 nm predicted by the modeling [46], which display the same hexagonal pattern with a central core as in the model in Figure 3. When the control starlet **F*** lacking complementarity to the template was used, the outcome was drastically different, with almost no discrete objects observed apart from what looked like isolated templates with no shells and a lot of unstructured material, likely resulting from uncontrolled polymerization of **F*** in the absence of the template.

We did not yet carry out the experiments on the detachment of the shell from the template by a DTT-mediated reduction in the disulfide linker as it was carried out previously [47] and the potential separation of the latter from the former. However, we are designing a FRET-based dual fluorophore assay to confirm the disassembly of the components of a complex in Figure 3 for a future study.

To conclude, following the initial molecular modeling of a C_24_ fullerene-like DNA hollow shell, we have synthesized the branched DNA and RNA oligonucleotides for the dendritic template and the branched *starlet* unit required for a template-assisted assembly of the above shell, carried out the assembly of the shell in a complex with the template, and verified the results by transmission electron microscopy (TEM). The obtained TEM images in the case of a complementary starlet revealed the formation of discrete nanoparticles, the size and morphology of which conform to the predictions of the modeling according to the concept of template-assisted assembly, whereas in the case of a non-complementary control starlet, no such objects were observed. The results can be useful for the design of more complex nucleic acid nanoarchitectures such as nanodevices and nanorobots.

## 4. Materials and Methods

General. Reverse-phased (RP) HPLC was carried out using acetonitrile (Supergradient UHPLC grade, Panreac, Madrid, Spain) was used. A 2 M solution of triethylammonium acetate (TEAA), pH 7.0, was prepared from triethylamine (ACS grade, Panreac, Madrid, Spain) and high purity glacial acetic acid (SoyuzKhimProm, Novosibirsk, Russia). Dichloroacetic acid, iodine, 0.25 M solution of 4,5-dicyanoimidazole (DCI) in anhydrous acetonitrile, Stains-All, Xylene Cyanol FF, and Bromophenol Blue (BP) dyes were from Sigma-Aldrich (Saint Louis, MO, USA), sodium perchlorate from Acros Organics (Carlsbad, CA, USA), dichloromethane, tetrahydrofuran, pyridine, and triethylamine from Panreac (Madrid, Spain). Formamide, acrylamide, *N*,*N*’-methylene-bis-acrylamide, urea, tris(hydroxymethyl)aminomethane (Tris), boric acid, a disodium salt of ethylenediaminetetraacetic acid (Na_2_EDTA) were from Dia-M (Moscow, Russia). Conc. aq. ammonia solution, acetic acid, and acetone, all of “high purity’ grade” were from SoyuzKhimProm (Novosibirsk, Russia). All reagents were of the highest purity offered by the respective manufacturers. Acetonitrile (Supergradient UHPLC grade, Panreac, Madrid, Spain) for oligonucleotide synthesis was refluxed for 6 h over CaH_2_ under argon atmosphere, distilled under argon, and stored under argon over 3 Å molecular sieves (Acros Organics, Carlsbad, CA, USA). Bi-distilled water was prepared in the laboratory. For centrifugation of small volumes of solutions, a MiniSpin Plus microcentrifuge (Eppendorf, Hamburg, Germany) is used. Chemical reactions are carried out using a Thermomixer Compact thermoshaker (Eppendorf, Hamburg, Germany). The solutions were shaken using a BioVortex V1 vortex (Biosan, Riga, Latvia). Gel electrophoresis is carried out using an electrophoresis unit from Bio-Rad (Hercules, CA, USA). Small volumes of oligonucleotide solutions up to 1.5 mL were evaporated in a Concentrator Plus vacuum concentrator (Eppendorf, Hamburg, Germany). Oligonucleotide solutions, after purification, were lyophilized using a FreeZone freeze-drier (Labconco, Kansas City, MO, USA).

Computer modeling. The models of DNA and RNA duplexes were created in the Avogadro software version 2.8.0 [63]. The obtained geometrical parameters were used for building and visualization of the models in the Blender software version 2.82. In-house scripts with bpy and bmesh modules were employed.

Synthesis, purification, and analysis of branched oligonucleotides. Oligonucleotides were obtained using a DNA/RNA synthesizer, Mermade MM-12, according to modified protocols of phosphoramidite synthesis on a scale of 1 μmol from the corresponding 5′-DMTr-3′-β-cyanoethyl-*N*,*N*-diisopropyl phosphoramidites of 2′-deoxy-, 2′-*O*-TBDMS-ribo-, and 2′-*O*-methylribonucleosides (Sigma-Aldrich, Saint Louis, MO, USA), as well as the “reversed” 3’-DMTr-5′-β-cyanoethyl-*N*,*N*-diisopropyl phosphoramidites of 2′-deoxy- and 2′-*O*-TBDMS-ribonucleosides (ChemGenes, Wilmington, MA, USA), and the corresponding controlled pore glass (CPG) polymer supports of 1000 Å pore size with grafted 2′-deoxy-, 2′-ribo-, or 2′-*O*-methylribonucleosides (Sigma-Aldrich, Saint Louis, MO, USA, or Link Technologies, Bellshill, UK). Commercially available non-nucleosidic branching β-cyanoethyl-*N*,*N*-diisopropyl phosphoramidites symmetric doubler (Glen Research, Sterling, VA, USA, Cat. No. 10-1920) and trebler (Glen Research, Sterling, VA, USA, Cat. No. 10-1922) were employed to introduce 3-way or 4-way junctions, respectively. Disulfide phosphoramidite C6 S-S modifier (Glen Research, Sterling, VA, USA, Cat. No. 10-1936) was used to introduce the disulfide cleavable linkage into the starlet **F** or **F*** (Figure 1). NPOM Caged dT phosphoramidite (Glen Research, Sterling, VA, USA, Cat. No. 10-1534) furnished the photolabile nucleotide unit. All phosphoramidites were dissolved in dry acetonitrile to a concentration of either 0.1 M or 0.15 M for doubler, trebler, and C6 S-S modifier. The coupling times varied from 30 s for deoxynucleoside 3′-phosphoramidites to 6 min for the “reversed” deoxynucleoside 5′-phosphoramidites, 2′-*O*-methylribonucleoside phosphoramidites, and the C6 S-S modifier, 10 min for 2′-*O*-TBDMS-ribonucleotide phosphoramidites, and 30 min for doubler and trebler.

For analytical HPLC, an Agilent 1220 chromatographic system (Agilent Technologies, Santa Clara, CA, USA) with UV detection at 260 nm and a ZORBAX Eclipse XDB-C18 5 µm 4.6 × 150 mm column (Agilent Technologies, Santa Clara, CA, USA) was used. The elution was carried out in a gradient of acetonitrile in 20 mM TEAA, pH 7.0 from 0 to 60% in 30 min, and a flow rate of 1 mL/min.

Oligonucleotides were synthesized without retaining the 5′-DMTr group (‘DMTr Off’ mode) followed by isolation by preparative polyacrylamide gel electrophoresis (PAGE) in 2–3 mm thick 20% gel under denaturing conditions (8 M urea) and desalting on a NAP-25 column with Sephadex G-25 (GE Healthcare, Buckinghamshire, UK) in the form of sodium salts. To control the quality of oligonucleotides, analytical electrophoresis was performed in 0.4 mm thick 20% gel under similar conditions: acrylamide—*N*,*N*′-methylene-bis-acrylamide (30:1), 8 M urea, 90 mM Tris-borate, pH 8.3, 2 mM Na_2_EDTA at a voltage of 50 V/cm (see Appendix A). Oligonucleotides were applied in a solution containing 8 M urea, 0.05% Xylene Cyanol FF, and 0.05% Bromophenol Blue. Bands were visualized by staining the gel with a solution of Stains-All dye (500 mg/L) in formamide, followed by rinsing with distilled water.

The concentrations of oligonucleotides were calculated from the optical densities of the solutions at 260 nm using an NP80Touch UV-Vis spectrophotometer (Implen, Munich, Germany).

The molecular masses of oligonucleotides were determined using ion-pair (IP) LC-MS with electrospray ionization (ESI) on a Bruker microOTOF-Q II mass spectrometer with a Bruker Elute SP LC liquid chromatography system (Bruker Daltonics, Leipzig, Germany) at a flow rate of 0.3 mL/min and a temperature of 45 °C, eluent A: 10 mM diisopropylamine, 15 mM hexafluoroisopropanol (HFIP) in deionized water; eluent B: 10 mM diisopropylamine, 15 mM HFIP, 20% milliQ water, 80% acetonitrile (UHPLC grade) with a step gradient of 0–1 min 100% A, 1–3.5 min 100% B in negative ion detection mode. Oligonucleotide samples were dissolved in an aqueous buffer containing 20 mM TEAA and 60% acetonitrile to a concentration of 0.1 mM. The volume of the analyzed sample was 10 μL. The instrument was calibrated using Bruker calibration standards and a set of oligodeoxynucleotides with known masses. The molecular masses of the oligonucleotides were calculated using sets of experimental m/z values determined for each analyzed sample (Appendix A). Profiles of IP-LC-MS analyses of the oligonucleotides **A**, **B**, **C**, and **F** are shown in Appendix A.

Template-assisted assembly. For the assembly of the dendron **D** (Figure 2), solutions of branched oligonucleotides **A**, **B**, and **C** (Appendix A) were mixed in a stoichiometric ratio of 6:3:1 in 1 × Tris-acetate (TAE) buffer, 12.5 mM MgCl_2_ to final concentrations of [**A**] 2.4 μM, [**B**] 1.2 μM, and [**C**] 0.4 μM, assuming the resulting concentration of the desired product **D** to be 0.4 μM. The final volume was 100 μL.

To assemble the 24-valent template **E**, a mixture of oligonucleotides **A**, **B**, and **C** was heated at 90 °C for 5 min with slow cooling to room temperature for 2 days in two different ways, namely, in a water bath with a cooling rate of approximately 0.1 °C/min, and in a Biosan dry thermostat, turning off the thermostat immediately after the incubation time at 90 °C. After incubation, a 40 µL aliquot of the solution was transferred into a 200 µL UV-transparent Eppendorf tube, followed by irradiation with a mercury lamp at a wavelength of 365 nm for 5 h to obtain the dimeric template **E**. After irradiation, the solution of the template was left for another 24 h at room temperature.

The assembly of a C_24_ fullerene-like DNA shell on the template **E** was carried out by adding the solution of the starlet **F** or a control non-complementary starlet **F*** (Appendix A) to the solution of the template from the previous step to the final concentration of 4.8 μM, assuming the concentration of the dimeric template **F** to be 0.2 μM. Next, the solutions were similarly irradiated for 7 h with UV light at 365 nm and left for 24 h at room temperature, after which the resulting samples were analyzed by native agarose gel electrophoresis and transmission electron microscopy (TEM).

Transmission electron microscopy. Visualization of the DNA/RNA nanostructures obtained by template-assisted assembly was carried out in a transmission electron microscope (TEM) after processing by the negative staining method. Briefly, the sample was adsorbed onto copper grids with a Formvar substrate for 1 min. Excess liquid was removed with filter paper, and the grid was placed on a drop of 1% aqueous uranyl acetate solution for 15 s. The excess of the contrast agent was drawn off with filter paper; the grids were dried in air. The preparations were studied in a JEM-1400 TEM (JEOL, Tokyo, Japan) at an accelerating voltage of 80 kV. The images were obtained with a Veleta side entry digital camera (EM SIS, Stuttgart, Germany).

## Figures and Tables

**Figure 1 ijms-24-15978-f001:**
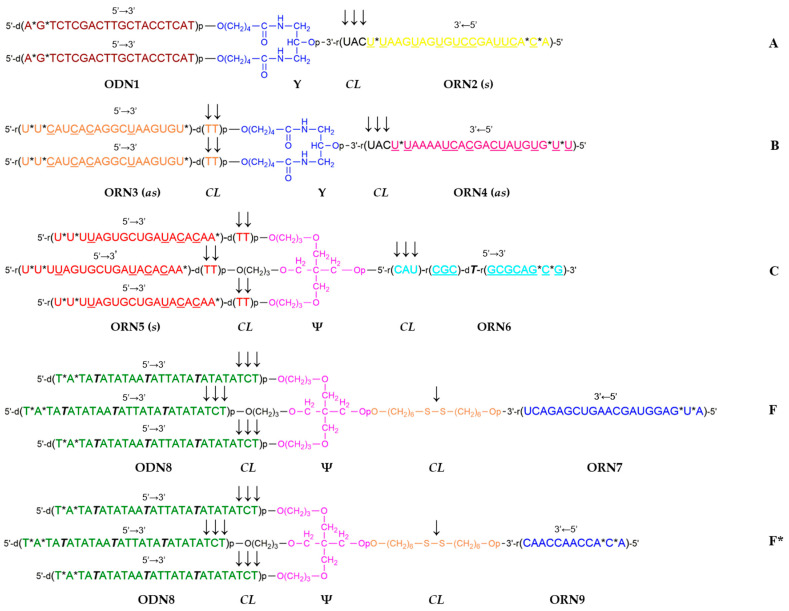
Structures of the branched oligonucleotides **A**, **B**, **C**, **F** (starlet), and **F*** (control non-complementary starlet). Notation: Y—symmetric doubler residue; Ψ—trebler residue; p—phosphodiester group [~P(=O)(O^−^)~]; *—phosphorothioate group [~P(=S)(O^−^)~]; ODN—oligodeoxynucleotide; ORN—oligoribonucleotide; *CL*—cleavable linker; ***T***—*N*^3^-(1-(2-nitropiperonyl)ethoxymethyl) (NPOM) thymidine; horizontal arrows indicate the direction of oligonucleotide sequences (5′-3′ or 3′-5′); vertical arrows indicate embedded biologically cleavable linkers (*CLs*) such as nuclease sensitive tri-ribonucleotides, tri-2′-deoxynucleotides (triple arrow), di-2′-deoxyribonucleotides (double arrow), or a disulfide bond (single arrow); marks (*s*) or (*as*) correspond to sense or antisense strands of siRNAs, respectively; deoxyribonucleotides and ribonucleotides are indicated by the prefixes *d* and *r*, respectively; 2′-*O*-methylribonucleotides are underlined; internucleotidic phosphodiester groups within sequences are omitted; coloring scheme (with the exception of **F** and **F***) is the same as in Figure 2.

**Figure 2 ijms-24-15978-f002:**
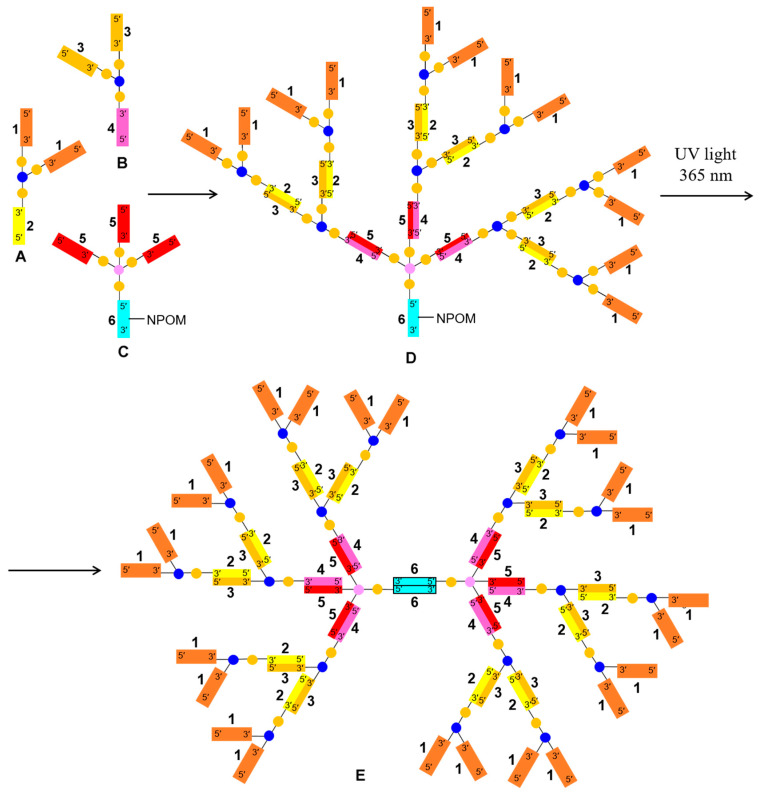
A scheme for the proposed assembly of a branched DNA/RNA template **E** by hybridization of branched oligonucleotides **A**, **B**, and **C** (Figure 1) to first give the dendron **D**, and ultimately, after UV irradiation at 365 nm, the dendrimer **E**. Key: Individual single strands **1**–**6** are represented as colored rectangles with their respective polarities marked (either 3′-5′ or 5′-3′); duplexes are shown as twin colored rectangles retaining the colors of their constituent single strands, e.g., dual colors for duplexes **2**:**3** and **4**:**5**, or a uniform color for the central duplex **6**:**6**. Blue circles mark symmetric doubler residues, lavender circles—trebler residues, orange circles—cleavable linkers (in **E**, some of the cleavable linkers are omitted for clarity). The color scheme follows Figure 1. NPOM—1-(2-nitropiperonyl)-ethoxymethyl group.

**Figure 4 ijms-24-15978-f004:**
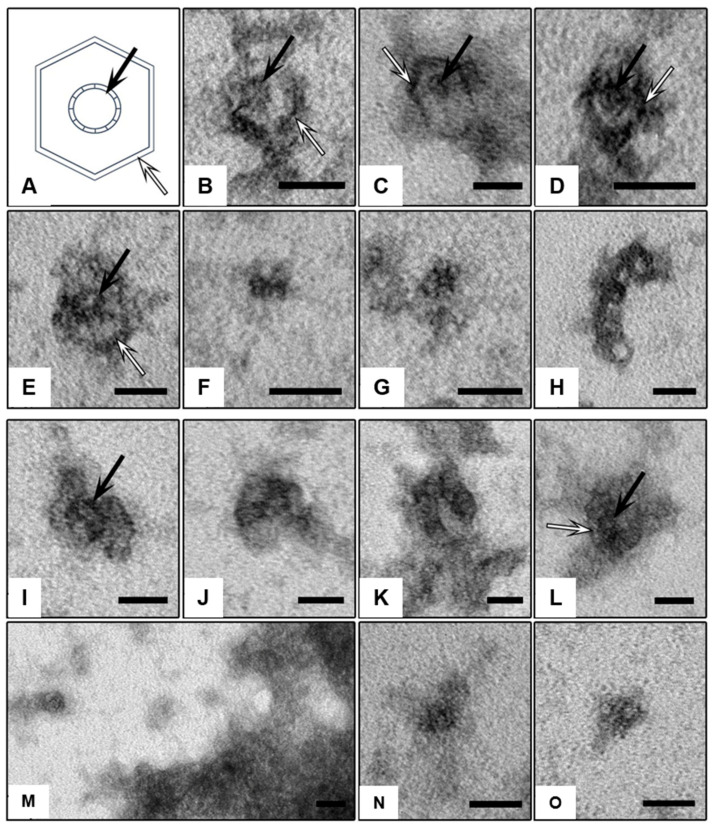
Representative images illustrating nanoparticles formed from branched oligonucleotides **E** and **F** obtained on a water bath (**B**–**E**) and in a dry thermostat (**F**–**I**). (**A**)—a schematic of the shell-and-core objects presented in the electron diffraction patterns. Particles corresponding to the isolated templates **E** or truncated shells: (**J**–**L**)—preparation obtained on a water bath; (**M**–**O**)—in a dry thermostat. Black arrows indicate the cores (probably shrunk templates) inside the shells, and white arrows indicate the “rods” in the shells. TEM, negative staining with 1% uranyl acetate solution. Scale bars correspond to 25 nm.

**Table 1 ijms-24-15978-t001:** Single oligoribonucleotide strands of siRNA duplexes. Adapted with permission from Ref. [46]. Copyright Pleiades Publishing, Ltd., 2019.

Description	Sequence	Code ^1^
msGankyrin-4 [57]	Sense (*s*) 3′-U*UAAGUAGUGUCCGAUUCA*C*A-5′ Antisense (*as*) 5′-U*U*CAUCACAGGCUAAGUGU*tt-3′	ORN2 ORN3
msPARP-1-4 [58]	Sense (*s*) 3′-U*UAAAAUCACGACUAUGUG*U*U-5′ Antisense (*as*) 5′-U*U*UUAGUGCUGAUACACAA*tt-3′	ORN4 ORN5

^1^ As in Figure 2. Notation: lowercase letters (t) denote deoxyribonucleotides; uppercase letters (A, U, G, and C) denote ribonucleotides; 2′-*O*-methylribonucleotides are underlined; *—phosphorothioate linkage.

## Data Availability

Data sharing not applicable.

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
