# Peer review of "Template-Assisted Assembly of Hybrid DNA/RNA Nanostructures Using Branched Oligodeoxy- and Oligoribonucleotides"

_ijms, 2023, doi:10.3390/ijms242115978_

Round 1

Reviewer 1 Report

Comments and Suggestions for Authors

The overall concept of building a DNA nanocage in the fashion described is suitably novel, although similar DNA nanostructures have been extensively reported elsewhere. However the lack of detailed data provision and analysis in the manuscript makes it hard to assess the results claimed. 

The gel elctrophoresis data must be shown, (the authors have in previous publications shown some gel data), to show the assembly has been successful after each step. 

The HPLC data provided in the SI, are all very similar for the range of DNA sequences, and have very short retention times. Were a range of elution gradients tested? Could other oligos be co-injected to indicate that the products have the indicated structures? Or do the authors expect all the oligos (despite being of a range of MWs ) to have the same retention times?

Are the authors ablr to provide any explanations for the small, but significant for G*, differences in the observed and calculated MS data? Is there a any evidence of common salt, or protonation site etc Some comment here would be important.

Although admirable, the TEM data is hard to intepret at the resolution provided. Fig 4, B, C, D and E do seem to show the expected structures however the other data is more difficult to appreciate. 

The use of Blender seems appropriate, however the introductory statements to its use are a bit of a tangent and do not add the to manuscript. Also the color coding between the model in Fig 3 and the schematic representations in Fig 1 and 2 are not similiar enough to make it easy to follow the assembly process. Fig 1 and 2 are also rather small, the dots in particular to be easily understood. 

Overall, the idea is suitably interesting to report, but the work lacks convincing data and needs to be more clearly presented if it is to be considered for publication.

Comments on the Quality of English Language

The majority of the work is described suitably, however there are some rather long sentences which are hard to follow and the introduction needs clarification. 

Author Response

Reviewer 1

  1. “The overall concept of building a DNA nanocage in the fashion described is suitably novel, although similar DNA nanostructures have been extensively reported elsewhere. However, the lack of detailed data provision and analysis in the manuscript makes it hard to assess the results claimed.”

We thank the Reviewer for commenting on the novelty of our approach. We tried to describe the results obtained in the most concise way; thus, the format of a short communication was chosen.

  1. “The gel electrophoresis data must be shown, (the authors have in previous publications shown some gel data), to show the assembly has been successful after each step.”

We have included a description of gel electrophoresis results into the revised Supporting Information (Fig. S2 and its description). We have also re-drawn Fig. 2 and re-named the oligonucleotides A–F and F* to better reflect the fact that the assembly was carried out in two stages only, and not three as it was shown previously. Thus, the oligonucleotide C was not actually made separately. We apologize for the confusion.

  1. “The HPLC data provided in the SI, are all very similar for the range of DNA sequences, and have very short retention times. Were a range of elution gradients tested? Could other oligos be co-injected to indicate that the products have the indicated structures? Or do the authors expect all the oligos (despite being of a range of MWs) to have the same retention times?”

We apologize for the confusion. The data shown in the Supporting Information (Fig. S1) relate to LC-MS analyses of the oligonucleotides. The conditions of the LC runs were standard; hence, the retention times were similar. We used preparative 15% denaturing (8 M urea) PAGE rather than HPLC to isolate and purify all the oligonucleotides. A representative analytical gel for a starlet branched unit has been included as an example (see Fig. S2 of the Supplementary Material).

  1. “Are the authors able to provide any explanations for the small, but significant for G*, differences in the observed and calculated MS data? Is there any evidence of common salt, or protonation site etc. Some comment here would be important.”

At the moment, we have no other explanation for the observed differences in the calculated and experimental MS data as the experimental error, which is understandably significant for MS analyses of oligonucleotides with MW in excess of 30 kDa (Table S1). No attempt was made yet to look for the evidence of salts, protonation etc. which may be possible for oligonucleotides with such a complex structure. We intend to carry out a thorough analysis at a later point, while studying biological effects of the obtained molecules.

  1. “Although admirable, the TEM data is hard to interpret at the resolution provided. Fig 4, B, C, D and E do seem to show the expected structures however the other data is more difficult to appreciate.”

We thank the Reviewer for the high rating of our TEM images. We tried to illustrate the most important elements of the construct. Unfortunately, our microscope, although very good, is nevertheless limited in its capabilities, and we were unable to further improve the images. Thus, we believe the resolution obtained is the best possible under the conditions available. We interpret the structures shown in Fig. 4B–E to represent the expected structure (Fig. 3), possibly, mixed with closely linked topologies with a different assortment of 6-membered, 5-membered, or 4-membered rings. Other dense objects with not so clearly defined structure have been interpreted as isolated templates or, possibly, incomplete assemblies (Fig. 4F–L), and, in some cases, as unstructured material (Fig. 4M).

  1. “The use of Blender seems appropriate, however the introductory statements to its use are a bit of a tangent and do not add to the manuscript. Also the color coding between the model in Fig 3 and the schematic representations in Fig 1 and 2 are not similiar enough to make it easy to follow the assembly process. Fig 1 and 2 are also rather small, the dots in particular to be easily understood.”

We modified the part of the Introduction related to Blender to make it more appropriate. Fig. 1 was intended as a 2-column figure, thus, it probably cannot be enlarged more than it is. However, we slightly increased the font for nucleotides to make the reading clearer. We have also completely re-drawn Fig. 2 to enlarge its elements and improve its visual perception. As to the colors that do not quite match between Fig. 3 and Figs. 1 and 2, it is due to the difference in the color schemes in the corresponding software packages (Blender and MS PowerPoint correspondingly). The duplexes represented as rectangles in Fig. 2 maintain dual colors of their respective single strands, e.g. 2:3 or 4:5. In Fig. 3, the duplexes represented as cylinders are colored uniformly with a single color only, which was noted in the legend to Fig. 3. In this sense the colors do not quite match indeed, and we removed the reference to the matching color scheme of Figs. 1 and 2 from the legend to Fig. 3.

  1. “Overall, the idea is suitably interesting to report, but the work lacks convincing data and needs to be more clearly presented if it is to be considered for publication.”

We thank the Reviewer for his or her interest in the concept of the work, and we did our best to improve the clarity of the paper as the Reviewer has requested. However, we respectfully disagree that the work “lacks convincing data”. We believe the obtained electron microscopy images confirm that the concept of template-assisted assembly is working as the change from the complementary starlet to a non-complementary starlet resulted in a complete disappearance of the objects that may be plausibly interpreted as the assemblies related to Fig. 3. Additionally, the size of the objects obtained (Fig. 4B–E) conforms nicely to the predicted by simulation (see also our earlier paper Bakulina et al, 2019 - ref. [46] in the uncorrected manuscript). One can also draw some important conclusions related to the topology of the objects from the obtained TEM pictures. We agree that more experiments need to be carried out to accumulate additional evidence, but we also believe that a short communication summarizing the work already done may be in our case both appropriate and timely.

  1. “The majority of the work is described suitably, however there are some rather long sentences which are hard to follow and the introduction needs clarification.”

We have re-written parts of Introduction avoiding long sentences as the Reviewer has pointed out, hopefully, improving the overall representation of the work.

Reviewer 2 Report

Comments and Suggestions for Authors

Dear Authors,

the present manuscript Template-Assisted Assembly of Hybrid DNA/RNA Nanostructures Using Branched Oligodeoxy- and Oligoribonucleotides by Alesya Fokina et al. describes the experimental preparation and characterization of supramolecular nucleic-acid assemblies in the form of fullerene-like structure. The current work stems from previous experience of the authors with more simple complexes (tetrahedrons and cubes) made in the same way and from the theoretical models published by the authors previously.

Major issues

  1. The introduction uses large parts (directly or modestly modified) from Fokina et al., 2021 [47] by a similar author team, e.g. „template-directed assembly of branched oligonucleotide blocks to form the vertices and edges of a ‘DNAhedron’ using a specifically synthesized branched oligonucleotide as a template (or a 3D scaffold), the structure of which is dictated by the shape of the target polyhedron, e.g., tetrahedron, cube, octahedron, etc.“ (l. 71-74) or “work has focused on solving two main tasks. Firstly, branched oligonucleotides for oligonucleotide templates for the assembly of a DNA tetrahedron and a DNA cube, and the branched oligonucleotide building block (‘starlet’) common for both polyhedra” (l. 86-89) or at other places. The introduction should be rewritten to avoid self-plagiarism and to focus more specifically on the current manuscript

  2. Without proper acknowledgment or reference, Fig. 2, Fig. 3, and Table 1 are improved versions of what appeared in Bakulina et al., 2019 [46] by a similar author team. This is also the case of the text “We hypothesized that the complex of DNA shell with the hybrid DNA/RNA template may exert biological activity if the internal duplexes of the template, namely duplexes 2:3 and 4:5 (Figure 2), represent small interfering RNAs (siRNAs)” (l. 140-142) and other small parts. Although the current paper is experimental and [46] is theoretical, the similarity and differences should be clearly stated (as foreseen by “The synthesis of branched oligonucleotides and the results of the assembly of nanostructures will be published later” in the end of “Results and Discussion” section of [46]).

  3. The final structure shown by the model in Fig. 3 (and in Fig. 9 of [46]) is not precisely that of (C24-Oh)[4,6] fullerene, where only squares and regular hexagons should appear. Instead, several pentagons formed by 8:8 duplexes are seen. Although this is noted on line 95 that it is only “topologically related to C 24 fullerene” and similarly on l. 100-101, it should be clarified at other places of the manuscript as well, especially in the caption of Fig. 3 (quotes in ‘C24 fullerene’ on l. 170 are not sufficient). Explanation as on l. 264-267 should come earlier. Better visualisation in Fig. 3, different aspects, more structured that can be formed, or a simplified diagram of the topologies would be also helpful.

  4. Outcomes of preparative polyacrylamide gel electrophoresis and analytical electrophoresis of the oligonucleotide samples (A, B, D, G, G*) and native agarose gel electrophoresis of the final complex should be shown .

  5. Did the authors experimentally check the correct assembly and structures of intermediate complexes C, E and F before proceeding to hybridization with G or G*?

  6. Can the author’s suggest further experiments to prove the existence of the proposed “fullerene” structures, such as fluorescence or EPR (using two fluorescent probes or radical labels, respectively)?

  7. Did the authors actually prepare a hollow shell by cleaving the template, as indicated by “a hollow C24 DNA shell may be constructed” on l. 230-231?

Minor comments

  1. In Fig. 2, labels i, ii, and iii should be explained or omitted since they are not referred to in other parts of the manuscript.

  2. l. 127-134 are not needed since they refer to the method used and the main conclusions were described already in [46]

  3. l. 184-218 describing TEM observations use words ‘core’ and ‘sphere’. If they correspond to the template and shell of the structures, the authors should keep this terminology. If ‘core’ and ‘sphere’ do note fully correspond to template and shell, it should be explained

  4. l. 247-248 state that “confirmation of the structure of the branched oligonucleotides (Figure 1) were carried out”. How exactly?

  5. Empty parentheses on l. 274

  6. The part Author contributions states that “The first three authors (A.F., S.D., and Y.P.) have contributed equally to the project” (l. 408-409), while only A.F. and Y.P. are marked by # in the header of the manuscript.

    Conclusion

    Because of the similarity with previously published works by the authors, major revisions are required to clearly differentiate the work that forms the current manuscript and express unambiguously its novelty.

Comments on the Quality of English Language

The authors use an excess of quoted words (starlets, sticky ends etc.), sometimes even italicized. This results into a rather too informal impression and the reader can doubt the scientific precision of the manuscript.

Author Response

Reviewer 2

  1. “The introduction uses large parts (directly or modestly modified) from Fokina et al., 2021 [47] by a similar author team, … The introduction should be rewritten to avoid self-plagiarism and to focus more specifically on the current manuscript”

            We have re-written the Introduction accordingly (see highlighted in the revised manuscript).

  1. “Without proper acknowledgment or reference, Fig. 2, Fig. 3, and Table 1 are improved versions of what appeared in Bakulina et al., 2019 [46] by a similar author team. This is also the case of the text “We hypothesized that the complex of DNA shell with the hybrid DNA/RNA template may exert biological activity if the internal duplexes of the template, namely duplexes 2:3 and 4:5 (Figure 2), represent small interfering RNAs (siRNAs)” (l. 140-142) and other small parts. Although the current paper is experimental and [46] is theoretical, the similarity and differences should be clearly stated (as foreseen by “The synthesis of branched oligonucleotides and the results of the assembly of nanostructures will be published later” in the end of “Results and Discussion” section of [46]).”

            Fig. 2 has been re-drawn to better reflect the actual experimental approach adopted in this work as compared to the theoretical concept outlined in Bakulina et al, 2019. Fig. 3 is not, in fact, the improved version of the Fig. 9 in Bakulina et al, 2019 as the topology of the shell is not the same (see also next comment). We think that the Table 1 is relevant to the Results section, and we would like to keep its current version. The reference to Bakulina et al, 2019 has been added as appropriate.

  1. “The final structure shown by the model in Fig. 3 (and in Fig. 9 of [46]) is not precisely that of (C24-Oh)[4,6] fullerene, where only squares and regular hexagons should appear. Instead, several pentagons formed by 8:8 duplexes are seen. Although this is noted on line 95 that it is only “topologically related to C 24 fullerene” and similarly on l. 100-101, it should be clarified at other places of the manuscript as well, especially in the caption of Fig. 3 (quotes in ‘C24 fullerene’ on l. 170 are not sufficient). Explanation as on l. 264-267 should come earlier. Better visualisation in Fig. 3, different aspects, more structured that can be formed, or a simplified diagram of the topologies would be also helpful.”

We have referred to C24 fullerene as a prototypical molecule, referring, in particular, to (C24-Oh)[4,6] fullerene with four- and six-membered rings as an example. However, there is another C24 fullerene isomer, namely, C24 (D6d) with 12 five- and 2 six-membered rings, which may be more directly related to the structure shown in Fig. 3 (see, e.g. Silant’ev A.V. Phys. Metals Metallogr. 121, 195–201 (2020), doi: 10.1134/S0031918X20010160). Thus, the isomer C24 (Oh) that we mention in the text was probably not particularly good as an example, and we thank the Reviewer for pointing it out to us. We have anticipated that our experiments may lead to different topological variants of a C24 fullerene-like DNA shell. As we think, the TEM data obtained (as seen in Fig. 4B–E) seem to better support the structure of C24 (D6d) for the shell, which is consistent with our optimized prediction shown in Fig. 3. However, it is clearly evident that further studies are needed to prove if the objects seen in TEM pictures are actually single isomers or the mixtures of topological isomers. The changes as per the Reviewer’s suggestion (p. 3) were introduced in the text.

  1. “Outcomes of preparative polyacrylamide gel electrophoresis and analytical electrophoresis of the oligonucleotide samples (A, B, D, G, G*) and native agarose gel electrophoresis of the final complex should be shown.”

            We added gel electrophoresis results to the Supplementary Material (Figs. S2 and S3).

  1. “Did the authors experimentally check the correct assembly and structures of intermediate complexes C, E and F before proceeding to hybridization with G or G*?”

            We have corrected Fig. 2 to reflect that the assembly of F (E in the revised manuscript as the naming for the oligonucleotides has been changed) was not stepwise but all three components were mixed simultaneously to form E and then F (after UV irradiation). Thus, we did not check the formation of the intermediate C. However, we have yet to devise a reliable technique to distinguish between partially assembled complexes (Figure S3). Nevertheless, after UV irradiation of the mixture of template F with starlet G, the fluorescence of ethidium bromide was much increased, which reflects the formation of new DNA duplexes in the shell. A similar increase was seen also in the case of non-complementary G*, yet no discrete objects were seen in TEM images as opposed to the correct starlet G.

  1. “Can the authors suggest further experiments to prove the existence of the proposed “fullerene” structures, such as fluorescence or EPR (using two fluorescent probes or radical labels, respectively)?”

            Yes, we have anticipated a FRET-based dual fluorophore experiment to confirm the detachment of the shell from the template in the future. A note referring to such an experiment was introduced in the Discussion.

  1. “Did the authors actually prepare a hollow shell by cleaving the template, as indicated by “a hollow C24 DNA shell may be constructed” on l. 230-231?”

            No, we did not yet perform the experiments to fragment and remove the template from the shell although the experiments with DTT reduction of the disulfide bridge were carried out previously to confirm if the concept is valid (ref [47] of the uncorrected manuscript). The experiments of DTT reduction are planned at the later stage of the project to try to isolate and characterize the hollow DNA shell free of template.

Minor comments

  1. “In Fig. 2, labels i, ii, and iii should be explained or omitted since they are not referred to in other parts of the manuscript.”

            The Fig. 2 was completely re-drawn, and the labels i, ii, and iii were removed.

  1. “l. 127-134 are not needed since they refer to the method used and the main conclusions were described already in [46]”

            The lines were removed.

  1. “l. 184-218 describing TEM observations use words ‘core’ and ‘sphere’. If they correspond to the template and shell of the structures, the authors should keep this terminology. If ‘core’ and ‘sphere’ do note fully correspond to template and shell, it should be explained”

            Corrected. Only the terms ‘template’ and ‘shell’ were used.

  1. “l. 247-248 state that “confirmation of the structure of the branched oligonucleotides (Figure 1) were carried out”. How exactly?”

            We used electrospray mass spectrometry (ESI MS) to confirm the molecular masses of the oligonucleotides (Table S1). That was mentioned in the Materials and Methods section (lines 371-372 of the original uncorrected version).

  1. “Empty parentheses on l. 274”

            Removed.

  1. “The part Author contributions states that “The first three authors (A.F., S.D., and Y.P.) have contributed equally to the project” (l. 408-409), while only A.F. and Y.P. are marked by # in the header of the manuscript.”

            Corrected to include only two first authors as required by the Journal rules.

  1. “The authors use an excess of quoted words (starlets, sticky ends etc.), sometimes even italicized. This results into a rather too informal impression and the reader can doubt the scientific precision of the manuscript.”

            We apologize for using the quoted words freely. To implement the Reviewer’s suggestion, we removed quotes (and italics) whenever appropriate. We believe that the words ‘sticky ends’ is a common slang term, which is used frequently in the molecular biology literature and is pretty well known to the wider readership. Thus, we employed it to designate specifically the single-stranded oligodeoxynucleotides 1 (Fig. 2) in the outer layer of the branched template F. The starlet (little star) is the term coined by us previously (see refs [46] and [47] of the uncorrected manuscript) for the branched unit of the outer shell, which has tetrahedral symmetry. As it was used in earlier papers, we would rather prefer to stick to it in the current manuscript.

Round 2

Reviewer 1 Report

Comments and Suggestions for Authors

The overall structure and presentation has been hugely improved and the paper reads much better - more or less ready for publication. The addition of gel data in the SI and the additional comments made in response to reveiwer questions have been completed satisfactorily and the work should be accepted for publication. 

Comments on the Quality of English Language

Some minor errors need checking, should be picked up by final proof readers of the journal.

Author Response

Reviewer 1

We thank the Reviewer 1 for his or her helpful comments.

As no more changes are requested apart from minor spell check, no specific changes have been necessary.

The updated version of the manuscript will be uploaded after proofreading, and any error will be corrected wherever found.

Reviewer 2 Report

Comments and Suggestions for Authors

Dear Authors,

thank you for addressing all the issues raised in my review and explaining the parts that were not clear to me. 

The manuscript has improve significantly.

Regarding the quoted words, I appreciate the reduction of the quotes and italics. I agree with the authors that the words starlets and sticky ends are, also in my view, quite common in the field. Therefore, I didn't mean to remove these words themselves but to reduce their emphasis, which is fine now (as matter of style, quotes around sticky ends could also be avoided).

There is only one remaining question: in your response, it is stated that a mention of a future FRET-based dual fluorophore experiment to confirm the detachment of the shell from the template in the future was introduced in the Discussion. However, I can't find this in the revised version. Please check this and update if necessary.

Author Response

Reviewer 2

We thank the Reviewer 2 for his or her very helpful suggestions.

As requested, quotes around sticky ends were removed in the updated version.

We apologize for missing the statement on a future FRET-based experiment in the Discussion. It has now been added in the appropriate section.